# Federated Auto-Meta-Ensemble Learning Framework for AI-Enabled Military Operations

**Konstantinos Demertzis** [1,*], **Panayotis Kikiras** [2], **Charalabos Skianis** [3], **Konstantinos Rantos** [4], **Lazaros Iliadis** [5] and **George Stamoulis** [2]

1 School of Science & Technology, Informatics Studies, Hellenic Open University, 26335 Patra, Greece
2 Department of Electrical and Computer Engineering, University of Thessaly, 38446 Volos, Greece
3 School of Engineering, Department of Information and Communication Systems Engineering, University of Aegean, 83200 Samos, Greece
4 School of Science, Department of Computer Science, International Hellenic University, 65404 Kavala, Greece
5 Department of Civil Engineering, School of Engineering, Democritus University of Thrace, 67100 Xanthi, Greece
* Correspondence: demertzis.konstantinos@ac.eap.gr

**Abstract:** One of the promises of AI in the military domain that seems to guarantee its adoption is its broad applicability. In a military context, the potential for AI is present in all operational domains (i.e., land, sea, air, space, and cyber-space) and all levels of warfare (i.e., political, strategic, operational, and tactical). However, despite the potential, the convergence between needs and AI technological advances is still not optimal, especially in supervised machine learning for military applications. Training supervised machine learning models requires a large amount of up-to-date data, often unavailable or difficult to produce by one organization. An excellent way to tackle this challenge is federated learning by designing a data pipeline collaboratively. This mechanism is based on implementing a single universal model for all users, trained using decentralized data. Furthermore, this federated model ensures the privacy and protection of sensitive information managed by each entity. However, this process raises severe objections to the effectiveness and generalizability of the universal federated model. Usually, each machine learning algorithm shows sensitivity in managing the available data and revealing the complex relationships that characterize them, so the forecast has some severe biases. This paper proposes a holistic federated learning approach to address the above problem. It is a Federated Auto-Meta-Ensemble Learning (FAMEL) framework. FAMEL, for each user of the federation, automatically creates the most appropriate algorithm with the optimal hyperparameters that apply to the available data in its possession. The optimal model of each federal user is used to create an ensemble learning model. Hence, each user has an up-to-date, highly accurate model without exposing personal data in the federation. As it turns out experimentally, this ensemble model offers better predictability and stability. Its overall behavior smoothens noise while reducing the risk of a wrong choice resulting from under-sampling.

**Keywords:** federated learning; model-agnostic; meta-learning; ensemble learning; military operations; cyber defense

## 1. Introduction

With an increasing pace, artificial intelligence (AI) is becoming a significant and integral part of modern warfare because it offers new opportunities for the complete automation of large-scale infrastructure and the optimization of numerous defence or cyber-defence systems [1]. One of the promises of AI in the military domain [2] that seems to guarantee its adoption is its broad applicability. In a military context, the potential for AI is present in all operational domains (i.e., land, sea, air, space, and cyber-space) and all levels of warfare (i.e., political, strategic, operational, and tactical) [3]. Still, at the same time, the complexity is growing exponentially as the number of interconnected systems

involved in continuous interconnection and uninterrupted information exchange services expands in real-time [4]. From a generalized point of view, it can be said that AI will have a significant impact on the following missions:

1. Too fast missions with reaction times of seconds or less to be executed in high complexity (data, context, type of mission).
2. Missions with operation duration beyond human endurance or implying high operational (personnel) costs over a long period.
3. Missions involving an overwhelming complexity which requires agility and adaptation to evolutions in context and objectives.
4. Missions challenging operational contexts implying severe risks to war fighters.

Applications supporting the missions above that monitor events in real-time are receiving a constant, unlimited stream of observations of interlinked approaches. These data exhibit high variability because their features vary substantially and unexpectedly over time, altering their typical, expected behaviour. The latest data are the most important in the typical case, as ageing is based on their timing.

Military AI-enabled intelligent systems that utilize data can transform military commanders' and operators' knowledge and experience into optimal valid and timely decisions [3,4]. However, the lack of detailed knowledge and expertise associated with using complex machine learning architectures can affect the performance of the intelligent model, prevent the periodic adjustment of some critical hyperparameters and ultimately reduce the algorithm's reliability and the generalization that should characterize these systems. These disadvantages are preventing stakeholders of defence, at all echelons of the command chain, from trusting and making effective and systematic use of machine learning systems. In this context and given the inability of traditional decision-making systems to adapt to the changing environment, the adoption of intelligent solutions is imperative.

Furthermore, a general difficulty that reinforces distrust of machine learning systems in defence is the prospect of adopting a single data warehouse for the overall training of intelligent models [1], which could create severe technical challenges and severe issues of privacy [5], logic, and physical security due to the need of establishing a potential single point of failure and a potential strategic/primary target for the adversaries [6]. Accordingly, the exchange of data that could make more complete intelligent categorizers that would generalize also poses risks to the security and privacy of sensitive data, which military commanders and operators do not want to risk [7].

To overcome the above double challenge, this work proposes FAMEL. It is a holistic system that automates selecting and using the most appropriate algorithmic hyperparameters that optimally solve a problem under consideration, approaching it as a model for finding algorithmic solutions where it is solved by mapping between input and output data. The proposed framework uses meta-learning to identify similar knowledge accumulated in the past to speed up the process [8]. This knowledge is combined using heuristic techniques, implementing a single, constantly updated intelligent framework. Data remains in the local environment of the operators, and only the parameters of the models are exchanged through secure processes, thus making it harder for potential adversaries to intervene with the system [9,10].

## 2. Proposed Framework

In the proposed FAMEL framework, each user uses an automatic meta-learning system in a horizontal federated learning approach (horizontal federated learning uses datasets with the same feature space across all devices. Vertical federated learning uses different datasets of different feature space to jointly train a global model). The most appropriate algorithm with the optimal hyperparameters is selected in a fully automated way, which can optimally solve the given problem. The implementation is based on the entity's available data and is not required to be disposed of in a remote repository or shared with a third party [11].

The whole process is described in Figure 1.

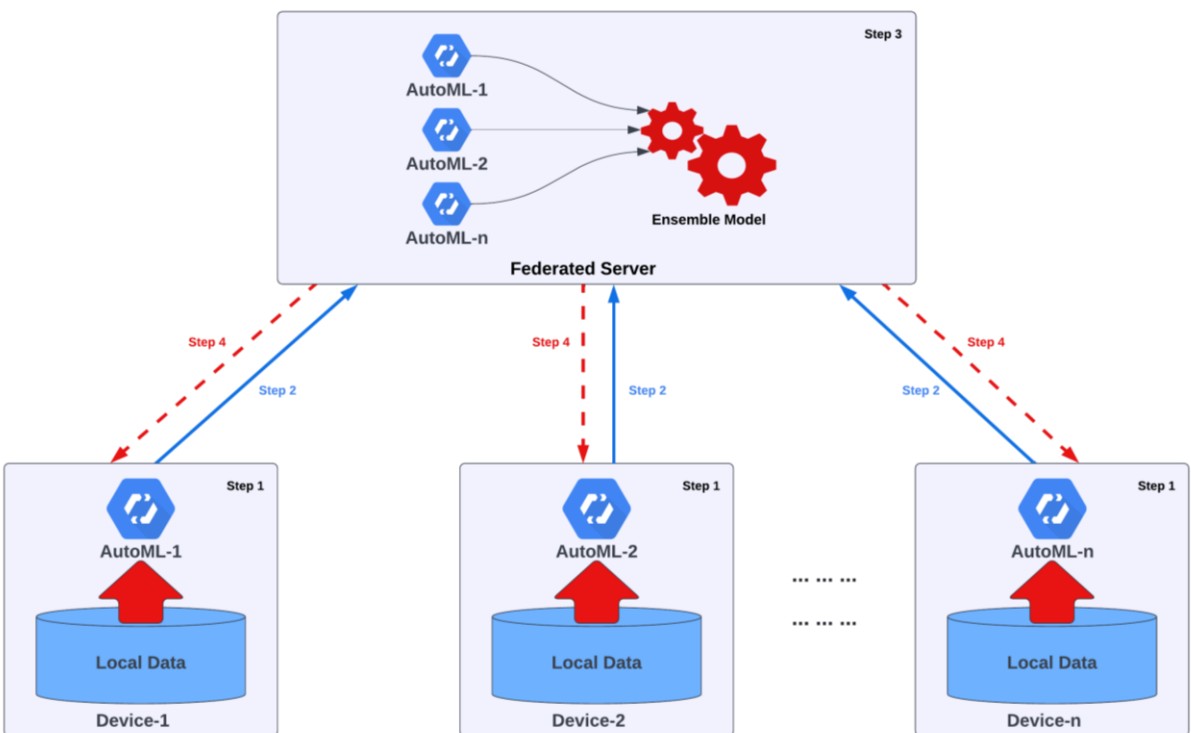

**Figure 1.** The proposed block diagram of the FAMEL framework.

Specifically:

5. Step 1—Fine-tune the best local model. The fine-tuning process will help to improve the accuracy of each machine learning model by integrating data from an existing dataset and using it as an initialization point to make the training process time- and resource-efficient.
6. Step 2—Upload the local model to the federated server.
7. Step 3—Ensemble the model by the federated server. This ensemble method uses multiple learning algorithms to obtain a better predictive performance than could be obtained from any of the constituent learning algorithms alone.
8. Step 4—Dispatch the ensemble model to local devices.

The best models (winner algorithm) that result from the process are channelled to a federated server, where an ensemble learning model through a heuristic mechanism is created. This ensemble model essentially incorporates the knowledge represented by the local best models, which, as mentioned, came from the local data held by the users [12]. Hence, collectively, the ensemble model offers high generalization, better predictability, and stability. Its general behaviour smoothens noise while lowering the overall danger of making a false choice due to modelling or prejudice in handling scenarios of local data [13,14].

### 2.1. Federated Learning

Assuming that $F_i = 1, 2, \ldots, N$ data owners want to train a machine learning model using their data $D = \{D_i, i = 1, 2, \ldots, N\}$. A traditional way would be to collect all data into a single set $D_{sum} = D_1 \cup D_1 \cup \cdots \cup D_N$ to train a model $M_{sum}$. The proposed federated learning system creates a single universal model [15]:

$$M_{fed} = \sum_{k=1}^{K} \frac{n_k}{n} w_{t+1}^k$$

where $K$ is the total number of nodes used in the process, $n$ is the data points, and t is the number of federated learning rounds [16]. The model comes from local models $\Delta w^1$, $\Delta w^2, \ldots, \Delta w^K$, which are trained from the $D_i$ of each federal user separately. Data D1 of the user F1 is not exposed to other federal users. In addition, the accuracy $V_{sum}$ and $M_{sum}$ of the models $V_{fed}$ and $M_{fed}$, must be close or equal. Specifically, if $\delta$ is a negative number, then the federal learning method suffers from a loss of accuracy, as indicated in the following formula [11,13]:

$$\left| V_{fed} - V_{sum} \right| < \delta$$

The Auto-Machine Learning technique is used to develop an accurate and robust federal system that will remain stable in new information without the ability to generalize or suffer a considerable loss of $\delta$-accuracy [17,18].

*2.2. Auto-Machine Learning*

Initially, each federation member has a set of $D$ data containing attribute vectors and class tags on a supervised problem linked with a job. Data set $D$ is specifically divided into two parts: a set of training S and a set of forecasts B for testing and testing so that $D = \langle S, B \rangle$. Furthermore, the data set $D$ contains vector-label pairings such that $D = \{(x_i, y_i)\}$. Each label represents a known class and belongs to a known set of $L$ labels [19].

Considering $P(D)$ the distribution of aggregate data held by federal agencies, we can sample the issuance of an individual data set such that $P_d = P_d(\mathbf{x}, y)$. Our problem lies in creating a trained classifier $\mathcal{M}_\lambda : \mathbf{x} \mapsto y$ which is fully and optimally configured with $\lambda \in \Lambda$ so that it can automatically generate predictions for samples from the $P_d$ distribution minimizing the expected generalization error so that [20]:

$$GE(\mathcal{M}_\lambda) = \mathbb{E}_{(\mathbf{x},y) \sim P_d}[\mathcal{L}(\mathcal{M}_\lambda(\mathbf{x}), y)]$$

The first phase is the best model selection procedure, which appears to be a standard learning procedure in which a data set is regarded as a sample of data. Furthermore, given that each data set of each independent body can only be observed through a set of $n$ independent observations, i.e.,:

$$D_d = \{(\mathbf{x}_1, y_1), \ldots, (\mathbf{x}_n, y_n)\} \sim P_d$$

Implies that we can only empirically approach the generalization error in data samples, i.e., [20,21]:

$$\widehat{GE}(\mathcal{M}_\lambda, \mathcal{D}_d) = \frac{1}{n} \sum_{i=1}^{n} \mathcal{L}(\mathcal{M}_\lambda(\mathbf{x}_i), y_i)$$

From the above, we conclude that we have access to unconnected, finite samples in practice where $D_{train}$ and $D_{test}$ ($D_{d,train}$, $D_{d,test} \in P_d$). Therefore, to search for the best machine learning algorithm, we only have access to $D_{train}$. However, in the end, the performance is calculated once in $D_{test}$.

Assume a classifier $f_\lambda$, the parameter $\lambda$ obtains the likelihood that a data point belongs to the class $y$ specified by the attribute vector $x$, $P_\lambda(y|x)$. The best model should increase the likelihood of correctly detecting tags over several training batches $B \subset D$ so that [18,22]:

$$\lambda^* = argmax_\lambda \mathbb{E}_{B \subset D} \left[ \sum_{(x,y) \in B} P_\lambda(y|x) \right]$$

Given that there is only a limited collection of quick learning support that can act as fine-tuning, the objective, as with any other work using machine learning, is to minimize the prediction error made on data samples with unknown labels. It is possible that obtaining the best model is challenging to undertake. A fake data set is created with only a tiny fraction of labels to prevent releasing all labels in the model. The optimization technique is modified to make it easier to acquire knowledge quickly. According to this interpretation, each sample

pair can be regarded as a data point. As a direct consequence, the model has been educated to the point where it can generalize to fresh, untested data sets. To summarize, the process of computing the best model through the application of the meta-learning approach is represented by the following function [20]:

$$\lambda^* = argmax_\lambda \mathbb{E}_{L_s \subset L} \left[ \mathbb{E}_{S^L \subset D, B^L \subset D} \left[ \sum_{(x,y) \in B^L} P_\lambda \left( x, y, S^L \right) \right] \right]$$

Therefore, the proposed framework performs an automatic search in the solutions area to identify the optimal $\mathcal{M}_{\lambda^*}$:

$$\mathcal{M}_{\lambda^*} \in \underset{\lambda \in \Lambda}{\operatorname{argmin}} \widehat{GE}(\mathcal{M}_\lambda, \mathcal{D}_{\text{train}})$$

For the calculation of GE, with cross-validation k-fold, the following relation is used [17,20,23]:

$$\widehat{GE}_{\text{CV}}(\mathcal{M}_\lambda, \mathcal{D}_{\text{train}}) = \frac{1}{K} \sum_{k=1}^{K} \widehat{GE} \left( \mathcal{M}_\lambda^{\mathcal{D}_{\text{train}}^{(\text{train},k)}}, \mathcal{D}_{\text{train}}^{(\text{val},k)} \right)$$

where $\mathcal{M}_\lambda^{\mathcal{D}_{\text{train}}^{(\text{train },k)}}$ denote that $\mathcal{M}_\lambda$ was trained based on the k-fold dataset $\mathcal{D}_{\text{train}}^{(\text{train },k)} \subset \mathcal{D}_{\text{train}}$ and then evaluated by:

$$\mathcal{D}_{\text{train}}^{(\text{val},k)} = \frac{\mathcal{D}_{\text{train}}}{\mathcal{D}_{\text{train}}^{(\text{train},k)}}$$

Accordingly, the problem of optimizing the hyperparameters $\lambda \in \Lambda$ of the best learning algorithm $A$ is essentially similar to selecting the best model. Some significant characteristics are that hyperparameters are frequently continuous, hyperparameter spaces are often vast, and we can benefit from the correlation between different hyperparameter settings $\lambda_1, \lambda_2, \ldots, \lambda_n \in \Lambda$.

Specifically, when $n$ hyperparameters $\lambda_1, \lambda_2, \ldots, \lambda_n \in \Lambda$ the hyperparameter space $L$ includes the subsets $\Lambda_1, \Lambda_2, \ldots, \Lambda_n$. This logic strictly defines each subset, so some hyperparameter settings make other hyperparameters inactive.

Specifically, the hyperparameter $\lambda_i$ is subject to the sub-constraints of another hyperparameter $\lambda_j$, if $\lambda$ is active only if the hyperparameter $\lambda_j$ takes values from a given set $V_i(j) \subsetneq \Lambda_j$. Based on this logic, the hyperparameters in the proposed framework create a structured solution space which is determined on the basis of a pair of variables with $B = \langle G, \Theta \rangle$ (where $G$ a graph). Graph $G$ conveys the assumption that each variable $\lambda_i$ is independent of the inheritance undertaken by $G$. It determines the parameters of the network and, in particular, the whole $\theta_{\lambda_i | \pi_i} = P_B(\lambda_i | \pi_i)$ for each $\lambda_i \subset \Lambda_i$ based on the constraint condition $\pi_i$, for the set of constraints in $G$. Therefore, $B$ defines a unique probability distribution such that [24]:

$$P_B = (\Lambda_1, \Lambda_2, \ldots, \Lambda_n) = \prod_{i=1}^{n} P_B(\pi_i) = \prod_{i=1}^{n} \theta_{\lambda_i} | \pi_i)$$

Finding the optimal graph path based on the Markov inequality is calculated as [25,26]:

$$\sum_{k=\omega}^{n} \binom{n}{k} (k-1)! p^k = \sum_{k=\omega}^{n} \frac{\prod_{i=0}^{k-1}(n-i)}{n^k} \frac{\lambda^k}{k} \le \sum_{k=\omega}^{n} \lambda^k = O(\lambda^\omega)$$

Hence, with the following equation, the calculation of its expectation is performed by $\Lambda_n$:

$$\mathbb{E}[\Lambda_n] = \sum_{k=3}^{n} \binom{n}{k} (k-1)! p^k$$

It follows from the above equation:

$$\lim_{n\to\infty} \mathbb{E}(\Lambda_n) = \lim_{n\to\infty} \sum_{k=3}^{n} \frac{\prod_{i=0}^{k-1}(n-i)}{n^k} \frac{\lambda^k}{k} \sim \sum_{k=3}^{\infty} \frac{\lambda^k}{k} = -\log(1-\lambda) - \lambda - \frac{\lambda^2}{2}$$
$$= a(\lambda)$$

Hence, its r-to factor momentum $\Lambda_n$ is:

$$\mathbb{E}[(\Lambda_n)_r] = \widehat{\theta} \sum_{k_1=3}^{n} \sum_{k_2=3}^{n-k_1} \cdots \sum_{k_r=3}^{n-\sum_{i=1}^{r-1}k_i} \binom{n}{k_1, k_2, \ldots, k_r, n-k_1-\cdots-k_r}$$
$$\prod_{i=1}^{r}(k_i-1)! p^{k_i}$$

Finally, given the above-structured solution space, the hyperparameter optimization issue is as follows [27,28]:

$$\lambda^* \in \operatorname{argmin}_{\lambda \in \Lambda} \frac{1}{k} \sum_{i=1}^{k} \mathcal{L}\left(A_\lambda, \mathcal{D}_{\text{train}}^{(i)}, \mathcal{D}_{\text{valid}}^{(i)}\right)$$

The Meta-Ensemble Learning technique is used for the proposed framework to lead to stable prediction models while offering generalization, minimizing bias, reducing variance, and eliminating overfitting.

### 2.3. Meta-Ensemble Learning

Once the above procedure has identified the most appropriate algorithm with the optimal hyperparameters to create a single model that improves generalization, the proposed framework creates a boosting ensemble model of all the optimal models that emerged by auto-machine learning.

The proposed technique is based on the logic of the boosting process, where through the creation of successive tree structures, information transfer is applied to solve a distributed problem [29]. Specifically, it is set $f(x) = 0$ and $\varepsilon_i = y_i$ for each observation in the set of training data of each body.

The winning algorithm from the process of auto-machine learning $\widehat{f^k}$ is trained in each round $k$ with $d$ nodes having as response variable the categorization errors resulting from the previous classification round (auto-machine learning process), which are denoted by $\varepsilon_i$. For the most efficient, effective, and computable feasible implementation of the proposed framework, we consider a tree and even pruned version of a new tree so that [12,30]:

$$\widehat{f}(x) \leftarrow \widehat{f}(x) + \lambda\widehat{f^k}(x) \text{ or } \varepsilon_i \leftarrow \varepsilon_i - \lambda\widehat{f^k}(x)$$

Repeating the procedure $K$ times (the user-specified $K$), the final form of the model is obtained:

$$\widehat{f}(x) = \lambda \sum_{k=1}^{K} \widehat{f^k}(x)$$

For the proposed technique to be effective, the user must specify the number and depth of trees to be created. The incredible depth of the trees can easily create over-adaptation processes and cannot be generalized. Accordingly, the number of trees controls the complexity of the process [31]. The $\lambda$ parameter defines the learning rate of the model.

Its derivative is first calculated to find the total minimum of the function using the proposed technique, and then the inverse procedure of finding the derivative is used. The derivative measures whether the value of a process will change $J(\theta)$ if the variable $\theta$ (slope of the function) changes slightly. High values of the function indicate a significant slope and, therefore, a substantial change in its value $J(\theta)$ for small changes of $\theta$.

This algorithm is iterative, initializes a random value in $\theta$, calculates the derivative of the function at the given point, and changes $\theta$ so that [28,32]:

$$\theta = \theta - \rho \frac{dj}{d\theta}$$

Taking as a function of loss the sum of the squares of the incorrect classifications $\varepsilon_i$ is divided by two so that: the parameter $\rho$ determines how fast it will move in the negative direction of the derivative. The process is repeated until the algorithm converges, which proposes training trees in the negative derivative of the loss function:

$$L(y_i, \widehat{y}_i) = \frac{1}{2} \sum_{i=1}^{N} (y_i - \widehat{y}_i)^2$$

Calculating the above derivative, we have:

$$\frac{dL(y_i, \widehat{y}_i)}{d\widehat{y}_i} = \widehat{y}_i - y_i$$

The negative derivative of the loss function is equal to the classification errors $\varepsilon_i$. Hence, essentially, the procedure provides for the training of a tree based on the classification errors $\varepsilon_i$, to which a pruned version of the new tree is added. In this manner, the approach adds successive trees to the negative derivative of the loss function at each given time $t$, such that [33,34]:

$$\widehat{y}_i^{(t)} = \sum_{t=1}^{K} f_t(x_i), \ f_t \in F$$

where $F = \left\{ f(x) = w_{q(x)} \right\}$ and $q : R^m \to T$, $w \in R^T$. The $q$ represents the structure of each tree, the $T$ the number of leaves, and each $f_t$ corresponds to an independent tree structure $q$ with the leaf weights plotted as $w$. The loss function that is minimized at any time $t$ has a formula:

$$L^{(t)} = \sum_{i=1}^{n} l\left(y_i, \widehat{y}_i^{t}\right) + \sum_{k=1}^{T} \Omega_{f(t)}$$

Two terms are important: the model's capacity for learning from training data (low values imply good learning) and the complexity of each tree (adding a new term to the number of leaves $(T)$, which shrinks the weights of leaves so that:

$$\Omega_{f(t)} = \gamma T + \frac{1}{2} \lambda \sum_{j=1}^{T} w_j^2$$

The parameter $\gamma$ indicates the penalty value for the tree's growth so that large values of $\gamma$ will lead to small trees. Respectively small values of $\gamma$ will lead to large trees. The parameter $\lambda$ regulates whether the tree weights will shrink so that as its value increases, the tree weights will shrink.

Thus, it follows that [33,35]:

$$\widehat{y}_i^{(t)} = \sum_{t=1}^{K} f_t(x_i) = \widehat{y}_i^{(t-1)} + f_t(x_i)$$

Therefore, the problem now is deciding which $f_t(x_i)$ minimizes the time loss function $t$:

$$L^{(t)} = \sum_{i=1}^{n} l\left(y_i, \widehat{y}_i^{(t)}\right) + \sum_{k=1}^{T} \Omega_{f(t)} = \sum_{i=1}^{n} l\left(y_i, \widehat{y}_i^{(t-1)} + f_t(x_i)\right) + \sum_{k=1}^{T} \Omega_{f(t)}$$

Taylor's Development shows:

$$f(x + \Delta x) \cong f(x) + f'(x)\Delta x + \frac{1}{2} f''(x)(\Delta x)^2$$

Hence, the resulting relationship is [33,35]:

$$L^{(t)} \cong \sum_{i=1}^{n} \left[ l\left(y_i, \widehat{y}_i^{(t-1)}\right) + g_i f_t(x_i) + \frac{1}{2} h_i f_t^2(x_i) \right] + \Omega_{f(t)}$$

where:

$$g_i = d_{\widehat{y}_i^{(t-1)}} l\left(y_i, \widehat{y}_i^{(t-1)}\right) \ and \ h_i = d_{\widehat{y}_i^{(t-1)}}^2 l\left(y_i, \widehat{y}_i^{(t-1)}\right)$$

Subtracting the constants, the loss function becomes:

$$L'^{(t)} \cong \sum_{i=1}^{n} \left[ g_i f_t(x_i) + \frac{1}{2} h_i f_t^2(x_i) \right] + \Omega_{f(t)}$$

where:

$$I_j = \{i | g(x_i) = j\}$$

The set of observations on a leaf $j$ in the above relation is recorded as follows:

$$L'^{(t)} \cong \sum_{i=1}^{n} \left[ g_i w_q(x_i) + \frac{1}{2} h_i w_q^2(x_i) \right] + \Omega_{f(t)} = \sum_{i=1}^{T} \left[ \left(\sum_{i \in I_j} g_i\right) w_j + \frac{1}{2}\left(\sum_{i \in I_j} h_i + \lambda\right) w_j^2 \right] + \gamma T$$

where:

$$G_j = \sum_{i \in I_j} g_i \ and \ H_j = \sum_{i \in I_j} h_i$$

The following relation emerges:

$$L'^{(t)} = \sum_{i=1}^{T} \left[ G_j w_j + \frac{1}{2}(H_j + \lambda) w_j^2 \right] + \gamma T$$

If the structure of the tree $(q(x))$ is given, the optimal weight on each sheet is obtained by minimizing the concerning $w_j$ in the above relationship so that [22,36,37]:

$$w_j = -\frac{G_j}{H_j + \lambda}$$

Finally, with its replacement $w_j$, the following equation is obtained, which calculates the quality of the new tree:

$$L'^{(t)} = -\frac{1}{2} \sum_{j=1}^{T} \frac{G_j^2}{H_j + \lambda} + \gamma T$$

Finally, the algorithm creates divisions using the formula:

$$Gain = \frac{1}{2} \left[ \frac{G_L^2}{H_L + \lambda} + \frac{G_R^2}{H_R + \lambda} - \frac{(G_L + G_R)^2}{H_L + H_R + \lambda} \right] - \gamma$$

where the first fraction is the score of the left part of the partition, the second is the score of the right amount of the division, the third is the score if the division is not made, and $\gamma$ measures the cost of the complexity of the partition.

## 3. Experiments and Results

For the experimental implementation of the proposed FAMEL and the performance of the scenario, a collaborative network of three federated partners (domain_alpha, domain_bravo and domain_charlie) was simulated (Figure 2).

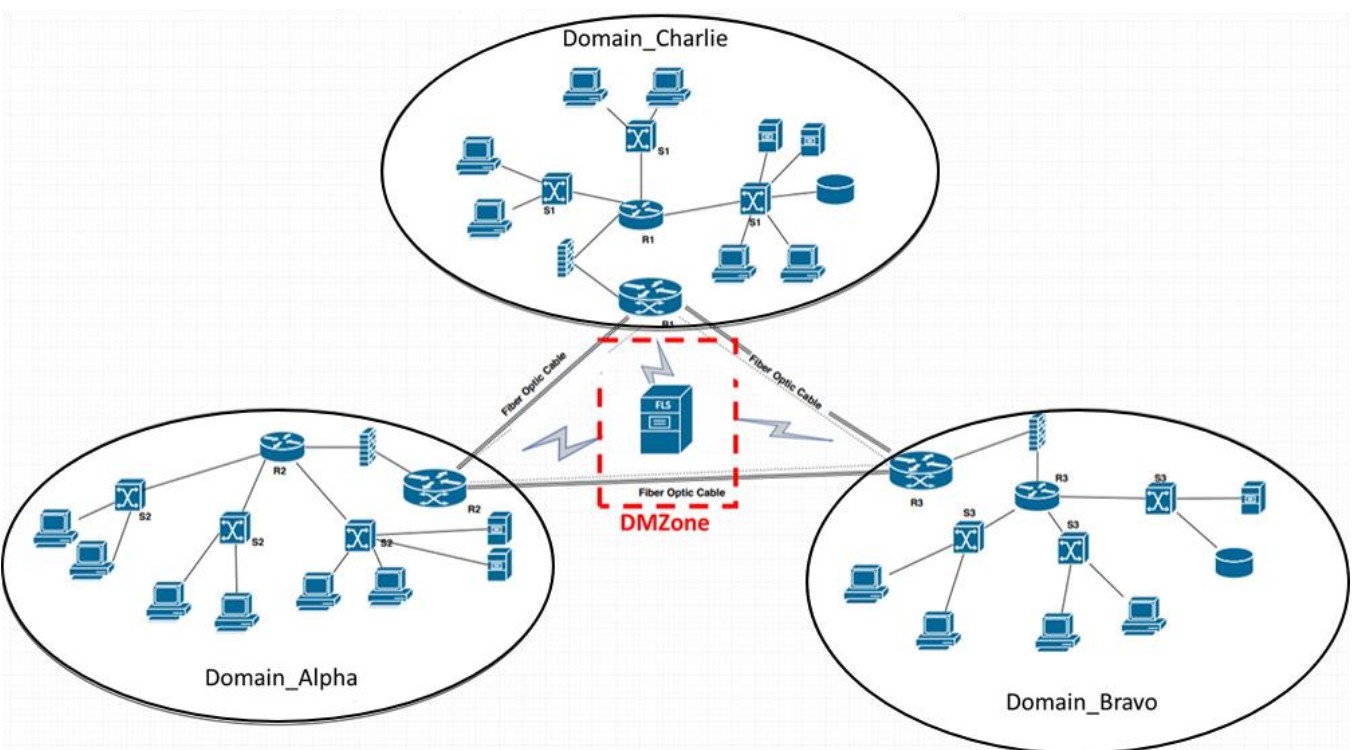

**Figure 2.** FAMEL architectural modelling.

We consider that the optimal model created by the Auto-Machine Learning process is an internal affair of each domain, which is implemented on a local server based on the respective architecture of each domain. In the Demilitarized Zone (DMZone) is the Federated Learning Server (LFS), which creates the ensemble model by applying the algorithmic process of assembling the optimal models with the technique discussed above. The proposed intelligent system was evaluated using one of the most extensive datasets for web traffic analysis called CICDoS2019. This dataset was developed under the supervision of the Canadian Institute for Cybersecurity. The evaluation's primary objective was to identify well-organized attacks in which the intruder's identity remained a legal component of a third party [31]. Each domain includes 70 independent variables: characteristics or statistics of network analysis and six classes (Benign, Infiltration, SSH-Bruteforce, FTP-BruteForce, DoS Attack-Hulk, and DDOS attack-HOIC). The individual sets include Alpha_dataset 70553, Bravo_dataset 69551, and Charlie_dataset 70128 instances [38].

The initial results of the Auto-Machine Learning process based on the data available in each domain are presented in Tables 1–9 below, as well as the parameters of each optimal model that emerged for each collaborative domain. We used the Area under the ROC Curve (AUC) metric, which represents the degree or measure of separability. It tells how much the model is capable of distinguishing between classes. Specifically, AUC (also known as AUROC) is the Area beneath the entire ROC curve. AUC provides a convenient, single performance metric for our classifiers independent of the specific classification threshold. This enables us to compare models without even looking at their ROC curves.

**Table 1.** The best model for the Domain Alpha.

| Domain Alpha | | | | | |
|---|---|---|---|---|---|
| **Model** | **Accuracy** | **AUC** | **Recall** | **Precision** | **F1-Score** |
| Light Gradient Boosting Machine | 0.879 | 0.926 | 0.876 | 0.879 | 0.879 |
| Gradient Boosting Classifier | 0.878 | 0.926 | 0.875 | 0.878 | 0.878 |
| K Neighbours Classifier | 0.876 | 0.927 | 0.873 | 0.876 | 0.876 |
| Logistic Regression | 0.873 | 0.924 | 0.869 | 0.873 | 0.873 |
| SVM—Linear Kernel | 0.870 | 0.925 | 0.867 | 0.870 | 0.870 |
| Ada Boost Classifier | 0.868 | 0.000 | 0.865 | 0.868 | 0.868 |
| Random Forest Classifier | 0.865 | 0.926 | 0.862 | 0.865 | 0.865 |
| Linear Discriminant Analysis | 0.864 | 0.924 | 0.861 | 0.864 | 0.864 |
| Ridge Classifier | 0.860 | 0.000 | 0.857 | 0.860 | 0.860 |
| Extra Trees Classifier | 0.853 | 0.920 | 0.852 | 0.853 | 0.853 |
| Decision Tree Classifier | 0.824 | 0.883 | 0.824 | 0.824 | 0.824 |
| Naive Bayes | 0.747 | 0.904 | 0.733 | 0.770 | 0.734 |
| Quadratic Discriminant Analysis | 0.367 | 0.900 | 0.405 | 0.575 | 0.321 |

**Table 2.** Best parameters of the winner model of the Domain Alpha.

| Domain_Alpha | |
|---|---|
| **Best Model** | **Best Parameters of the Winner Model** |
| LGBMClassifier | boosting_type = 'gbdt', class_weight = None, colsample_bytree = 1.0, importance_type = 'split', learning_rate = 0.1, max_depth = −1, min_child_samples = 20, min_child_weight = 0.001, min_split_gain = 0.0, n_estimators = 100, n_jobs = −1, num_leaves = 31, objective = None, random_state = 1599, reg_alpha = 0.0, reg_lambda = 0.0, silent = 'warn',subsample = 1.0, subsample_for_bin = 200,000, subsample_freq = 0 |

**Table 3.** The best model for the Domain Bravo.

| Domain_Bravo | | | | | |
|---|---|---|---|---|---|
| **Model** | **Accuracy** | **AUC** | **Recall** | **Precision** | **F1-Score** |
| Gradient Boosting Classifier | 0.877 | 0.926 | 0.875 | 0.877 | 0.877 |
| Light Gradient Boosting Machine | 0.876 | 0.926 | 0.874 | 0.876 | 0.876 |
| K Neighbours Classifier | 0.876 | 0.926 | 0.873 | 0.874 | 0.875 |
| Ada Boost Classifier | 0.870 | 0.925 | 0.868 | 0.870 | 0.870 |
| Random Forest Classifier | 0.870 | 0.923 | 0.868 | 0.870 | 0.870 |
| Linear Discriminant Analysis | 0.865 | 0.923 | 0.863 | 0.865 | 0.865 |
| SVM—Linear Kernel | 0.865 | 0.000 | 0.863 | 0.865 | 0.865 |
| Logistic Regression | 0.863 | 0.925 | 0.861 | 0.863 | 0.862 |
| Ridge Classifier | 0.861 | 0.000 | 0.859 | 0.862 | 0.861 |
| Extra Trees Classifier | 0.849 | 0.920 | 0.849 | 0.849 | 0.849 |
| Decision Tree Classifier | 0.816 | 0.878 | 0.816 | 0.816 | 0.815 |
| Naive Bayes | 0.739 | 0.905 | 0.727 | 0.765 | 0.724 |
| Quadratic Discriminant Analysis | 0.594 | 0.917 | 0.570 | 0.572 | 0.545 |

**Table 4.** Best parameters of the winner model of the Domain Bravo.

| Domain_Bravo | |
|---|---|
| **Best Model** | **Best Parameters of the Winner Model** |
| GradientBoostingClassifier | ccp_alpha = 0.0, criterion = 'friedman_mse', init = None, learning_rate = 0.1, loss = 'deviance', max_depth = 3, max_features = None, max_leaf_nodes = None, min_impurity_decrease = 0.0, min_impurity_split = None, min_samples_leaf = 1, min_samples_split = 2, min_weight_fraction_leaf = 0.0, n_estimators = 100, n_iter_no_change = None, presort = 'deprecated', random_state = 8515, subsample = 1.0, tol = 0.0001, validation_fraction = 0.1, verbose = 0, warm_start = False |

**Table 5.** The best model for the Domain Charlie.

| Domain_Charlie | | | | | |
|---|---|---|---|---|---|
| **Model** | **Accuracy** | **AUC** | **Recall** | **Precision** | **F1-Score** |
| k-Neighbours Classifier | 0.866 | 0.927 | 0.864 | 0.867 | 0.866 |
| Light Gradient Boosting Machine | 0.865 | 0.926 | 0.864 | 0.866 | 0.866 |
| Gradient Boosting Classifier | 0.865 | 0.926 | 0.865 | 0.865 | 0.866 |
| Ada Boost Classifier | 0.861 | 0.921 | 0.861 | 0.861 | 0.861 |
| Logistic Regression | 0.860 | 0.922 | 0.860 | 0.861 | 0.860 |
| SVM—Linear Kernel | 0.855 | 0.923 | 0.852 | 0.855 | 0.855 |
| Random Forest Classifier | 0.853 | 0.925 | 0.851 | 0.853 | 0.853 |
| Linear Discriminant Analysis | 0.851 | 0.923 | 0.849 | 0.852 | 0.851 |
| Extra Trees Classifier | 0.847 | 0.921 | 0.847 | 0.848 | 0.849 |
| Ridge Classifier | 0.847 | 0.920 | 0.848 | 0.849 | 0.848 |
| Decision Tree Classifier | 0.819 | 0.880 | 0.821 | 0.820 | 0.819 |
| Naive Bayes | 0.687 | 0.900 | 0.668 | 0.680 | 0.644 |
| Quadratic Discriminant Analysis | 0.542 | 0.914 | 0.536 | 0.662 | 0.528 |

**Table 6.** Best parameters of the winner model of the Domain Charlie.

| Domain_Charlie | |
|---|---|
| **Best Model** | **Best Parameters of the Winner Model** |
| KNeighborsClassifier | algorithm = 'auto', leaf_size = 30, metric = 'minkowski', metric_params = None, n_jobs = −1, n_neighbors = 5, $p$ = 2, weights = 'uniform' |

**Table 7.** Ensemble model for the Domain Alpha.

| Domain_Alpha | | | | | |
|---|---|---|---|---|---|
| **Model** | **Accuracy** | **AUC** | **Recall** | **Precision** | **F1-Score** |
| Ensemble model | 0.898 | 0.933 | 0.899 | 0.897 | 0.898 |
| Light Gradient Boosting Machine | 0.879 | 0.926 | 0.876 | 0.879 | 0.879 |
| Gradient Boosting Classifier | 0.878 | 0.926 | 0.875 | 0.878 | 0.878 |
| k-Neighbors Classifier | 0.876 | 0.927 | 0.873 | 0.876 | 0.876 |
| Logistic Regression | 0.873 | 0.924 | 0.869 | 0.873 | 0.873 |

**Table 7.** *Cont.*

| Domain_Alpha | | | | | |
| --- | --- | --- | --- | --- | --- |
| **Model** | **Accuracy** | **AUC** | **Recall** | **Precision** | **F1-Score** |
| SVM—Linear Kernel | 0.870 | 0.925 | 0.867 | 0.870 | 0.870 |
| Ada Boost Classifier | 0.868 | 0.000 | 0.865 | 0.868 | 0.868 |
| Random Forest Classifier | 0.865 | 0.926 | 0.862 | 0.865 | 0.865 |
| Linear Discriminant Analysis | 0.864 | 0.924 | 0.861 | 0.864 | 0.864 |
| Ridge Classifier | 0.860 | 0.000 | 0.857 | 0.860 | 0.860 |
| Extra Trees Classifier | 0.853 | 0.920 | 0.852 | 0.853 | 0.853 |
| Decision Tree Classifier | 0.824 | 0.883 | 0.824 | 0.824 | 0.824 |
| Naive Bayes | 0.747 | 0.904 | 0.733 | 0.770 | 0.734 |
| Quadratic Discriminant Analysis | 0.367 | 0.900 | 0.405 | 0.575 | 0.321 |

**Table 8.** Ensemble model for the Domain Bravo.

| Domain_Bravo | | | | | |
| --- | --- | --- | --- | --- | --- |
| **Model** | **Accuracy** | **AUC** | **Recall** | **Precision** | **F1-Score** |
| Ensemble model | 0.891 | 0.930 | 0.890 | 0.890 | 0.892 |
| Gradient Boosting Classifier | 0.877 | 0.926 | 0.875 | 0.877 | 0.877 |
| Light Gradient Boosting Machine | 0.876 | 0.926 | 0.874 | 0.876 | 0.876 |
| k-Neighbors Classifier | 0.876 | 0.926 | 0.873 | 0.874 | 0.875 |
| Ada Boost Classifier | 0.870 | 0.925 | 0.868 | 0.870 | 0.870 |
| Random Forest Classifier | 0.870 | 0.923 | 0.868 | 0.870 | 0.870 |
| Linear Discriminant Analysis | 0.865 | 0.923 | 0.863 | 0.865 | 0.865 |
| SVM—Linear Kernel | 0.865 | 0.000 | 0.863 | 0.865 | 0.865 |
| Logistic Regression | 0.863 | 0.925 | 0.861 | 0.863 | 0.862 |
| Ridge Classifier | 0.861 | 0.000 | 0.859 | 0.862 | 0.861 |
| Extra Trees Classifier | 0.849 | 0.920 | 0.849 | 0.849 | 0.849 |
| Decision Tree Classifier | 0.816 | 0.878 | 0.816 | 0.816 | 0.815 |
| Naive Bayes | 0.739 | 0.905 | 0.727 | 0.765 | 0.724 |
| Quadratic Discriminant Analysis | 0.594 | 0.917 | 0.570 | 0.572 | 0.545 |

**Table 9.** Ensemble model for the Domain Charlie.

| Domain_Charlie | | | | | |
| --- | --- | --- | --- | --- | --- |
| **Model** | **Accuracy** | **AUC** | **Recall** | **Precision** | **F1-Score** |
| Ensemble model | 0.871 | 0.929 | 0.871 | 0.871 | 0.872 |
| k-Neighbors Classifier | 0.866 | 0.927 | 0.864 | 0.867 | 0.866 |
| Light Gradient Boosting Machine | 0.865 | 0.926 | 0.864 | 0.866 | 0.866 |
| Gradient Boosting Classifier | 0.865 | 0.926 | 0.865 | 0.865 | 0.866 |
| Ada Boost Classifier | 0.861 | 0.921 | 0.861 | 0.861 | 0.861 |
| Logistic Regression | 0.860 | 0.922 | 0.860 | 0.861 | 0.860 |
| SVM—Linear Kernel | 0.855 | 0.923 | 0.852 | 0.855 | 0.855 |

**Table 9.** *Cont.*

| Domain_Charlie | | | | | |
|---|---|---|---|---|---|
| **Model** | **Accuracy** | **AUC** | **Recall** | **Precision** | **F1-Score** |
| Random Forest Classifier | 0.853 | 0.925 | 0.851 | 0.853 | 0.853 |
| Linear Discriminant Analysis | 0.851 | 0.923 | 0.849 | 0.852 | 0.851 |
| Extra Trees Classifier | 0.847 | 0.921 | 0.847 | 0.848 | 0.849 |
| Ridge Classifier | 0.847 | 0.920 | 0.848 | 0.849 | 0.848 |
| Decision Tree Classifier | 0.819 | 0.880 | 0.821 | 0.820 | 0.819 |
| Naive Bayes | 0.687 | 0.900 | 0.668 | 0.680 | 0.644 |
| Quadratic Discriminant Analysis | 0.542 | 0.914 | 0.536 | 0.662 | 0.528 |

AUC is measured on a scale of 0 to 1, with higher numbers indicating better performance. Scores in the [0.5, 1] range indicate good performance, while anything less than 0.5 indicates very poor performance. An AUC of 1 indicates a perfect classifier, while an AUC of 0.5 indicates a perfectly random classifier. A model that always predicts a negative sample is more likely than a positive sample to have a positive label. It will have an AUC of 0, indicating a severe modelling failure.

It should be noted that all the tests were performed with 10-fold cross-validation. Each of the ten subsets was used for the algorithm's training and certainly once for its evaluation, so there was no case of misleading the algorithmic result.

Meta-Ensemble Learning is created with an ensemble model that includes the best classifiers. The ensemble model returns through the Federated Learning process in each domain and retests in each local dataset (Alpha_dataset, Bravo_dataset, and Charlie_dataset). Then, the three best models from each domain (LGBMClassifier, Gradient BoostingClassifier, and k-NeighborsClassifier) are sent with the Federated Learning process to FLS. Again, it should be emphasized that all the tests were performed with the method of 10-fold cross-validation so that there was no case of misleading the algorithmic result. The results of the process are presented in the following tables.

The ensemble model ensures an improved categorization accuracy and smoothening of the system. This dramatically simplifies trend detection and visualization by eliminating or reducing statistical noise in the data. The experimental results suggest that using the ensemble model ensures improved categorization accuracy. The categorization becomes more accurate with each instance, providing critical pointers to the failure problems that an individual algorithm's bias could generate [39]. This allows for a precise diagnosis before embarking on a new condition or occurrence associated with adversarial attacks or zero-day exploits. This is one of the most effective strategies for predicting a trend's strength and the likelihood of shifting direction [40].

The convergence achieved by employing multiple models provides more outstanding reliability than any of them could provide separately. This revelation, directly related to the experimental outcomes, significantly accelerates arriving at the optimum decision in ambiguous situations [41]. It is also important to remember that this process is dynamic, which must be emphasized. This dynamic process ensures the system's adaptability by providing impartiality and generalization, resulting in a system that can respond to highly complicated events [42,43].

## 4. Conclusions

Applying machine learning to real-world problems is still particularly challenging [44]. This is because highly trained engineers and military specialists who have a wealth of experience and information will be required to coordinate the numerous parameters of the respective algorithms, correlate them with the specific problems, and use the data sets that are currently available. This is a lengthy, laborious, and expensive undertaking. However,

the hyperparametric features of algorithms and the design choices for ideal parameters can be viewed as optimization problems because machine learning can be thought of as a search problem that attempts to approach an unknown underlying mapping function between input and output data.

Utilizing the above view, in the present work, FAMEL was presented, extending the idea of formulating a general framework of automatic machine learning with effective universal optimization, which operates at the federal level. It uses automated machine learning to find the optimal local model in the data held by each federal user and then, making extensive meta-learning, creates an ensemble model, which, as shown experimentally, can generalize, providing highly reliable results. In this way, the federal bodies have a dedicated, highly generalized model, the training of which does not require exposure to the federation of the data in their possession. In this regard, FAMEL can be applied to several military applications where continuous learning and environmental adaptation are critical for the supported operations and where the exchange of information might be difficult or not possible due to security reasons. For example, which is the case in the real-time optimization of information sharing concerning tasks and situations. The application of FAMEL would be of special interest in congested environments where IoT sensor grids are deployed, and many security constraints need to be met. Similarly, it can be applied in cyberspace operations to find and identify potential hostile activities in cluttered information environments and complex physical scenarios in real-time, including countering negative digital influence [45,46]. It must be noted that the proposed technique can be extended to cover a wider scientific area without reducing the main points that are currently described. It is a universal technic that develops and produces an open-frame holistic federated learning approach.

Although, in general, the methodology of the federated learning technique, the ensemble models, and recently the meta-learning methods have occupied the research community intensely, and relevant work has been proposed that has upgraded the relevant research area, this is the first time that such a comprehensive framework is presented in the international literature. The methodology offered herein is an advanced form of learning. The computational process is not limited to solving a problem but through a productive method of searching the solution space and selecting the optimal one in a meta-heuristic way [47,48].

On the other hand, the federated learning model should apply average aggregation methods to the set of cooperative training data. This raises serious concerns for the effectiveness of this universal approach and, therefore, for the validity of federated architectures in general. Generally, it flattens the unique needs of individual users without considering the local events to be managed. How one can create personalized intelligent models that solve the above limitations is currently a prominent research problem. For example, the study [49] is based on the needs and events that each user must address in a federated format. Explanations are the assortment of characteristics of the interpretable system, which, in the case of a specified illustration, helped to bring about a conclusion and provided the function of the model on both local and global levels. Retraining is suggested only for those features for which the degree of change is considered quite important for the evolution of its functionality.

Essential topics that could expand the research area of the proposed framework concern the Meta-Ensemble Learning process and, specifically, how to solve the problem of creating trees and their depth so that the process is automatically fully simplified. An automated process should also be identified for pruning each tree with optimal separations to avoid negative gain. Finally, explore procedures to add an optimally trimmed tree version to the model to maximize frame efficiency, accuracy, and speed.

**Author Contributions:** Conceptualization, K.D. and P.K.; methodology, K.D., P.K. and K.R.; software, K.D.; validation, K.D., P.K., C.S., K.R., L.I. and G.S.; formal analysis, C.S., K.R. and L.I.; investigation, K.D. and P.K.; resources, C.S., K.R., L.I. and G.S.; data curation, K.D., P.K., C.S., K.R., L.I. and G.S.; writing—original draft preparation, K.D. and P.K.; writing—review and editing, K.D., P.K., C.S., K.R., L.I. and G.S.; visualization, K.D. and K.R.; supervision, P.K. and C.S.; project administration, L.I. and G.S.; funding acquisition, C.S. and G.S. All authors have read and agreed to the published version of the manuscript.

**Funding:** This research received no external funding.

**Institutional Review Board Statement:** Not applicable.

**Informed Consent Statement:** Not applicable.

**Data Availability Statement:** Data Availability https://www.unb.ca/cic/datasets/ddos-2019.html (accessed on 29 November 2022).

**Conflicts of Interest:** The authors declare no conflict of interest.

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
