# Peer review of "Federated Auto-Meta-Ensemble Learning Framework for AI-Enabled Military Operations"

_electronics, doi:10.3390/electronics12020430_

Round 1

Author Response

Dear respected Reviewer,

We deeply appreciate your time and effort in reviewing our manuscript. Your comments are very helpful for revising and improving our paper. We have revised the manuscript considering all the insightful comments to enhance the paper's readability. We believe these changes have strengthened the rationale and importance of our study.

Yours sincerely,

Konstantinos Demertzis, Panagiotis Kikiras, Charalabos Skianis, Konstantinos Rantos, Lazaros Iliadis, George Stamoulis

Review Report (Reviewer 1)

This paper proposed Federated Auto-Meta-Ensemble Learning (FAMEL) framework for Military Operations. The idea and application of this paper are interesting. However, there are a few points the authors must address.

  • Authors must add a paragraph explanation for existing work with tabular comparison.

ANS-1: Thank you for this helpful comment. Unfortunately, there isn't another comparable model to use as a benchmark. Consequently, to avoid bias or incorrect impressions, we present the performance of the proposed model without making any comparisons with any other alternative models.

  • The quality of the images is not good. Particularly, in figure 2 some texts are blur and are not readable.

ANS-2: Thank you for your insightful feedback. We have enhanced all figures in 300 dpi to increase the paper's readability.

  • The authors did not show the ROC curve of the models.

ANS-3: Thank you for this constructive comment. We have added detailed explanations of why do not use the ROC metric, specifically, “In this paper, we used the Area under the ROC Curve (AUC) metric, which represents the degree or measure of separability. It tells how much the model is capable of distinguishing between classes. Specifically, AUC (also known as AUROC) is the Area beneath the entire ROC curve. AUC provides a convenient, single performance metric for our classifiers independent of the specific classification threshold. This enables us to compare models without even looking at their ROC curves. AUC is measured on a scale of 0 to 1, with higher numbers indicating better performance. Scores in the [0.5, 1] range indicate good performance, while anything less than 0.5 indicates very poor performance. An AUC of 1 indicates a perfect classifier, while an AUC of 0.5 indicates a perfectly random classifier. A model that always predicts a negative sample is more likely than a positive sample to have a positive label. It will have an AUC of 0, indicating a severe modelling failure.”

  • What do you mean by n, k, and t in line 116? Number of individual model or some things else. Elaborate clearly.

ANS-4: Thank you for this constructive comment. We have added the following explanations “where K the total number of nodes used in the process, n data points and t the number of federated learning rounds [16].”

  • What are the limitations in the individual model and sum models?

ANS-5: Thank you for this helpful comment. We have added detailed explanations in the conclusion section. Specifically, “The federated learning model should apply average aggregation methods to the set of cooperative training data. This raises serious concerns for the effectiveness of this universal approach and, therefore, for the validity of federated architectures in general. Generally, it flattens the unique needs of individual users without considering the local events to be managed. How one can create personalized intelligent models that solve the above limitations is in the primary research area. For example, the study [48] is based on the needs and events that each user must address in a federated format. Explanations are the assortment of characteristics of the interpretable system, which, in the case of a specified illustration, helped to bring about a conclusion and provided the function of the model on both local and global levels. Retraining is suggested only for those features for which the degree of change is considered quite important for the evolution of its functionality.”

  • The paper is written in more generalized way, it should be specific to address particular problem.

ANS-6: Thank you for this comment. As mentioned in the introduction section “Military AI-enabled Intelligent systems utilizing data can transform military commanders' and operators' knowledge and experience into optimal valid and timely decisions [3], [4]. However, the lack of detailed knowledge and expertise associated with using complex machine learning architectures can affect the performance of the intelligent model, prevent the periodic adjustment of some critical hyperparameters and ultimately re-duce the algorithm's reliability and the generalization that should characterize these systems. These disadvantages prevent stakeholders of defence at all echelons of the command chain from trusting and making effective and systematic use of machine learning systems. In this context, and given the inability of traditional decision-making systems to adapt to the changing environment, the adoption of intelligent solutions is imperative.

Also, a general difficulty that reinforces distrust of machine learning systems is the prospect of adopting a single data warehouse for the overall training of intelligent models [1], which creates severe technical glitches and severe issues of privacy [5], logic, and physical security due to the need of establishing a potential single point of failure and a potential strategic/primary target for the adversaries [6]. Accordingly, the exchange of data that could make more complete intelligent categorizers that will generalize also poses risks to the security and privacy of sensitive data, which military commanders and operators do not want to risk [7].

To overcome the above double challenge, this work proposes FAMEL. It is a holistic system that automates selecting and using the most appropriate algorithmic hyperparameters that optimally solve a problem under consideration, approaching it as a model for finding algorithmic solutions where it is solved by mapping between input and output data. The proposed framework uses meta-learning to identify similar knowledge accumulated in the past to speed up the process [8]. This knowledge is combined using heuristic techniques, implementing a single, constantly updated intelligent framework. Data remains in the local environment of the operators, and only the parameters of the models are exchanged through secure processes, thus making it harder for potential adversaries to intervene with the system [9], [10].”

Also, we have added in the conclusion section, the following explanation “In the present work, FAMEL was presented, the logic of which extends the idea of formulating a general framework of automatic machine learning with effective universal optimization, which operates at the federal level. It uses auto-mated machine learning to find the optimal local model in the data held by each federal user and then, making extensive meta-learning, creates an ensemble model, which, as shown experimentally, can generalize, giving highly reliable results. In this way, the federal bodies have a dedicated, highly generalized model, the training of which does not require exposure to the federation of the data in their possession. In this regard, FAMEL can be applied to several military applications where continuous learning and environ-mental adaptation are critical for the supported operations. For example, in real-time optimization of information sharing concerning tasks and situations. The application of FAMEL would be of special interest in congested environments where IoT sensor grids are deployed and many security constraints need to be met. Similarly, it can be applied in cyberspace operations to find and identify potential hostile activities in cluttered information environments and complex physical scenarios in real-time, including countering negative digital influence [44], [45]. It must be noted that the proposed technique can be extended to cover a wider scientific area without reducing the main points that are currently described. It is a universal technic that develops and produces an open-frame holistic federated learning approach.”

  • If the models are trained with different number of features, then federated server send back best weighted models to each device so the testing data from each device is different then how it handles the problem?

ANS-7: Thank you for this constructive comment. As mentioned in the Proposed Framework section “In the proposed FAMEL framework, each user uses an automatic meta-learning system in a horizontal federated learning approach (horizontal federated learning uses datasets with the same feature space across all devices. Vertical federated learning uses different datasets of different feature space to jointly train a global model).”

  • References are not cited in proper way. References should be started from number 1 but it is starting from 4. Also check reference number 3 in line 47.

ANS-8: We corrected some mistakes and double-checked the number of references referenced in the study. Thank you for your input.

Reviewer 2 Report

-The paper should be interesting ;;;

-it is a good idea to add a block diagram of the proposed research/review (step by step);;;

-it is a good idea to add more photos of measurements, sensors + arrows/labels what is what  (if any);;;

-What is the result of the analysis?;;

-figures should have high quality. ;;;;;

-text should be formatted;;;;

-please add photos of the application of the proposed research, 2-3 photos ;;; 

-what will society have from the paper?;;

-labels of figures should be bigger;;;;

-Is there a possibility to use the proposed research for other topics;;;

-references should be from the web of science 2020-2022 (50% of all references, 30 references at least);;;

-Conclusion: point out what have you done;;;;

-please add some sentences about future work;;;

Author Response

Dear respected Reviewer,

We deeply appreciate your time and effort in reviewing our manuscript. Your comments are very helpful for revising and improving our paper. We have revised the manuscript considering all the insightful comments to enhance the paper's readability. We believe these changes have strengthened the rationale and importance of our study.

Yours sincerely,

Konstantinos Demertzis, Panagiotis Kikiras, Charalabos Skianis, Konstantinos Rantos, Lazaros Iliadis, George Stamoulis

Review Report (Reviewer 2)

The paper should be interesting.

-it is a good idea to add a block diagram of the proposed research/review (step by step).

ANS: Thank you for this constructive comment. The whole process and the individual parts of FAMEL, are described in detail in Figure 1

-it is a good idea to add more photos of measurements, sensors + arrows/labels what is what (if any).

ANS: Thank you for your suggestions. This research study is part of more extensive and long-term research published in stages. The first phase is only a non-GUI application. The second phase will deep into GUI applications and visualization in more technical details and reports. This is an overview article, and we aim to be a wide technical range to provide lasting value for the specific knowledge domain. We aim to give the readers a spherical overview of all fields of interest. After completing the second phase of this research, we aim to create an open-source repository for free use by the research community.

-What is the result of the analysis?

ANS: Thank you for this comment. The Experiments and Results section includes detailed explanations. Specifically, “The ensemble model ensures improved categorization accuracy, smoothing the system. This dramatically simplifies trend detection and visualization by eliminating or reducing statistical noise in the data. The experimental results suggest that using the ensemble model ensures improved categorization accuracy. The categorization becomes more accurate with each instance, providing critical pointers to the failure problems that an individual algorithm's bias could generate [38]. This allows for a precise diagnosis before embarking on a new condition or occurrence associated with adversarial attacks or zero-day exploits. This is one of the most effective strategies for predicting a trend's strength and the likelihood of shifting direction [39].

The convergence achieved by employing multiple models provides more outstanding reliability than any of them could provide separately. This revelation, directly related to the experiment outcomes, significantly accelerates arriving at the optimum decision in ambiguous situations [40]. It is also important to remember that this process is dynamic, which must be emphasized. This dynamic process ensures the system's adaptability by providing impartiality and generalization, resulting in a system that can respond to highly complicated events [41], [42].”

-figures should have high quality.

ANS: Thank you for your insightful feedback. We have enhanced all figures in 300 dpi to increase the paper's readability.

-text should be formatted.

ANS: Thank you for your constructive comment. We have revised the manuscript according to the journal format, considering your helpful comments to improve the paper's readability.

-please add photos of the application of the proposed research, 2-3 photos.

ANS: Thank you for your suggestions. This research study is part of more extensive and long-term research published in stages. The first phase is only a non-GUI application. The second phase will deep into GUI applications and visualization in more technical details and reports. This is an overview article, and we aim to be a wide technical range to provide lasting value for the specific knowledge domain. We aim to give the readers a spherical overview of all fields of interest. After completing the second phase of this research, we aim to create an open-source repository for free use by the research community.

-what will society have from the paper?

ANS: Thank you for this comment. We have added detailed explanations in the conclusion section. Specifically, “Applying machine learning to real-world problems is still particularly challenging [43]. This is because highly trained engineers and specialists who have a wealth of experience and information will be required to coordinate the numerous parameters of the respective algorithms, correlate them with the specific problems, and use the data sets that are currently available. This is a lengthy, laborious, and expensive undertaking. However, the hyperparametric features of algorithms and the design choices for ideal parameters can be viewed as optimization problems because machine learning can be thought of as a search problem that attempts to approach an unknown underlying mapping function between input and output data.

Utilizing the above view, in the present work, FAMEL was presented, the logic of which extends the idea of formulating a general framework of automatic machine learning with effective universal optimization, which operates at the federal level. It uses automated machine learning to find the optimal local model in the data held by each federal user and then, making extensive meta-learning, creates an ensemble model, which, as shown experimentally, can generalize, giving highly reliable results. In this way, the federal bodies have a dedicated, highly generalized model, the training of which does not require exposure to the federation of the data in their possession. In this regard, FAMEL can be applied to several research applications where continuous learning and adaptation are critical for the research community.”

-labels of figures should be bigger.

ANS: Thank you for this constructive comment. We have enhanced all figures considering your helpful suggestions to increase the figure's readability.

-Is there a possibility to use the proposed research for other topics.

ANS: Thank you for this comment. We have added in the conclusion section, the following explanation “It must be noted that the proposed technique can be extended to cover a wider scientific area without reducing the main points that are currently described. It is a universal technic that develops and produces an open-frame holistic federated learning approach.”

-references should be from the web of science 2020-2022 (50% of all references, 30 references at least).

ANS: Thank you for your careful reading. 31 of 48 references (65%) are between 2020 - 2022. Also, there are some substantial references from 2017 - 2020, and all others are crucial to the defence of this research study. 

-Conclusion: point out what have you done.

ANS: Thank you for this comment. The revised conclusion includes the following “in the present work, FAMEL was presented, the logic of which extends the idea of formulating a general framework of automatic machine learning with effective universal optimization, which operates at the federal level. It uses automated machine learning to find the optimal local model in the data held by each federal user and then, making extensive meta-learning, creates an ensemble model, which, as shown experimentally, can generalize, giving highly reliable results. In this way, the federal bodies have a dedicated, highly generalized model, the training of which does not require exposure to the federation of the data in their possession. In this regard, FAMEL can be applied to several military applications where continuous learning and environmental adaptation are critical for the supported operations. For example, in real-time optimization of information sharing concerning tasks and situations. The application of FAMEL would be of special interest in congested environments where IoT sensor grids are deployed, and many security constraints need to be met. Similarly, it can be applied in cyberspace operations to find and identify potential hostile activities in cluttered information environments and complex physical scenarios in real-time, including countering negative digital influence [44], [45].  It must be noted that the proposed technique can be extended to cover a wider scientific area without reducing the main points that are currently described. It is a universal technic that develops and produces an open-frame holistic federated learning approach.

Although, in general, the methodology of the federated learning technique, the ensemble models, and recently the meta-learning methods have occupied the research community intensely, and relevant works have been proposed that have upgraded the relevant research area, this is the first time that such a comprehensive framework is presented in the international literature. The methodology offered herein is an advanced form of learning. The computational process is not limited to solving a problem but through a productive method of searching the solution space and selecting the optimal one in the meta-heuristic way [46], [47].”

Also, added some limitations, specifically “On the other hand, the federated learning model should apply average aggregation methods to the set of cooperative training data. This raises serious concerns for the effectiveness of this universal approach and, therefore, for the validity of federated architectures in general. Generally, it flattens the unique needs of individual users without considering the local events to be managed. How one can create personalized intelligent models that solve the above limitations is in the primary research area. For example, the study [48] is based on the needs and events that each user must address in a federated format. Explanations are the assortment of characteristics of the interpretable system, which, in the case of a specified illustration, helped to bring about a conclusion and pro-vided the function of the model on both local and global levels. Retraining is suggested only for those features for which the degree of change is considered quite important for the evolution of its functionality.”

-please add some sentences about future work.

ANS: Thank you for this constructive comment. The revised conclusion includes the following “Essential topics that could expand the research area of the proposed framework concern the Meta-Ensemble Learning process and, specifically, how to solve the problem of creating trees and their depth so that the process is automatically fully simplified. An automated process should also be identified for pruning each tree with optimal separations to avoid negative Gain. Finally, explore procedures to add an optimally trimmed tree version to the model to maximize frame efficiency, accuracy, and speed.”

Round 2

Reviewer 2 Report

the block diagram of research should be added;;;

Author Response

Dear respected Reviewer,

We deeply appreciate your time and effort in reviewing our manuscript. Your comments are very helpful for revising and improving our paper. We have revised the manuscript considering all the insightful comments to enhance the paper's readability. We believe these changes have strengthened the rationale and importance of our study.

Yours sincerely,

Konstantinos Demertzis, Panagiotis Kikiras, Charalabos Skianis, Konstantinos Rantos, Lazaros Iliadis, George Stamoulis

Review Report (Reviewer 2)

-the block diagram of research should be added.

ANS: Thank you for this constructive comment. The proposed block diagram of the FAMEL framework depicted in Figure 1.

Figure 1. The proposed block diagram of the FAMEL framework

Also, we have added the following explanations

Specifically:

  1. Step 1 – Fine-tune the best local model. The fine-tuning process will help to improve the accuracy of each machine learning model by integrating data from an existing dataset and using it as an initialization point to make the training process time and resource-efficient.
  2. Step 2 – Upload the local model to the federated server.
  3. Step 3 – Ensemble the model by the federated server. This ensemble method uses multiple learning algorithms to obtain better predictive performance than could be obtained from any of the constituent learning algorithms alone.
  4. Step 4 – Dispatch the ensemble model to local devices.